# Bias Neutralization in Non-Parallel Texts:
# A Cyclic Approach with Auxiliary Guidance

**Karthic Madanagopal, James Caverlee**
Texas A&M University, College Station, USA
`karthic11@tamu.edu, caverlee@tamu.edu`

## Abstract

Objectivity is a goal for Wikipedia and many news sites, as well as a guiding principle of many large language models. Indeed, several methods have recently been developed for automatic subjective bias neutralization. These methods, however, typically rely on parallel text for training (i.e. a biased sentence coupled with a non-biased sentence), demonstrate poor transfer to new domains, and can lose important bias-independent context. Toward expanding the reach of bias neutralization, we propose in this paper a new approach called FairBalance. Three of its unique features are: i) a cycle consistent adversarial network enables bias neutralization without the need for parallel text; ii) the model design preserves bias-independent content; and iii) through auxiliary guidance, the model highlights sequences of bias-inducing words, yielding strong results in terms of bias neutralization quality. In our evaluations involving seven models comprising of adversarial and non-adversarial models, the FairBalance method showed a notable improvement in bias neutralization based on subjective human judgment when compared to other techniques.

## 1 Introduction

Avoiding bias is a cornerstone of many heavily relied upon resources. Examples include Wikipedia (Greenstein and Zhu, 2012), scholarly articles (Politzer-Ahles et al., 2020), many news sources (Liu et al., 2021), as well as emerging large language models (Patel and Pavlick, 2021). In particular, *subjective bias* is the use of biased language in presenting objective information with an implied proposition or conclusion (Wiebe, 1994). Biased language can manipulate our perception of reality and intensify social conflicts (Greenstein and Zhu, 2012, 2014; Beukeboom and Burgers, 2017).

While many communication venues strive for objectivity, subjective writing can still arise

**Biased Statements**

The "winners" are chosen by a group of academics, activists, **distinguished** **businessmen**, and trade unionists.

Released on May 16, 2002, Attack of the Clones was **generally perceived** as a **slight improvement** upon the **feeble** The Phantom Menace, **though not at all on par** with the original Star Wars trilogy.

**Only a tiny proportion** of these companies have **so far** grown into multinationals: ARM, Autonomy Corporation, and AVEVA are the **most obvious examples**, and more recently CSR has seen rapid growth due to the uptake of Bluetooth.

Table 1: Sample of biased statements extracted from Wikipedia. Spans highlighted are the output of our bias tagger by type such as **Epistemological Bias**, **Framing Bias** and **Demographic Bias**.

(Greenstein and Zhu, 2012; Liu et al., 2021; Recasens et al., 2013). To illustrate, Table 1 highlights sentences from Wikipedia displaying epistemological bias (i.e. casting a doubt), framing bias (i.e. one-sided frames), and demographic bias (i.e. unequal representation based on demographic characteristics). Encouragingly, a number of studies have identified efficient methods to detect biased language in text (Recasens et al., 2013; Bhosale et al., 2013; Misra and Basak, 2016; Hube and Fetahu, 2018; Zhong, 2021; Madanagopal and Caverlee, 2022) and a few studies have began to explore automatic subjective bias neutralization methods (Pryzant et al., 2019; Liu et al., 2021; Madanagopal and Caverlee, 2023). These approaches, however, face a number of key challenges:

**Reliance on Parallel Training Data.** The majority of existing subjective bias neutralization models rely on supervised methods trained over a

dataset of paired examples (Pryzant et al., 2019; Liu et al., 2021; Madanagopal and Caverlee, 2023), consisting of a biased sentence X (e.g., "John is a great actor") and its neutral version Y (e.g., "John is an actor"). These datasets are challenging and expensive to prepare, and simply do not exist for many domains.

**Inefficient Domain Adaptation.** Even with models trained in one domain, there is often poor adaptation to other domains. For example, words like "affordable health" or "abortion" may be highly charged in political speeches (Sim et al., 2013; Misra and Basak, 2016; Liu et al., 2021), but of little salience in other domains. Since Wikipedia is the primary training dataset for existing subjective bias neutralization methods (Pryzant et al., 2019; Zhong, 2021), patterns of bias there may not generalize.

**Content Preservation.** Besides neutralization, subjective bias correction aims to preserve the bias-independent information from the original text in the new neutral text. Existing methods use n-gram precision metrics like BLEU to refine the performance of content preservation by comparing the generated neutral text against human-generated reference text (Pryzant et al., 2019; Zhong, 2021). The absence of such reference (parallel) text motivates the need for training methods that naturally incorporate the content preservation objective.

To address these challenges, we propose Fair-Balance, a *content preserving subjective bias neutralization* approach that is designed to mitigate subjective bias in text without the need for parallel data. Our approach employs a cyclic adversarial network architecture, which includes two sequential generative adversarial networks (GAN) with a pre-trained discriminators. The first GAN takes a biased text as input and generates a neutral text and the second GAN takes the output of the first GAN as input to generate a biased text. By computing the cyclic loss between the original biased text with the generated biased text, the network ensures the bias independent content is preserved in the subjective neutralization process. Another key property of FairBalance is an auxiliary guidance mechanism with the help of a bias tagger that guides the generator toward identifying biased portions of the text and providing instructions on what should be eliminated or rephrased. By integrating this approach, we accelerate the training process and achieve more consistent and reliable

bias neutralization results.

In summary:

- We propose a cycle-consistent adversarial network for subjective bias neutralization, which can be trained effectively in the absence of parallel data and efficiently preserves the bias-independent characteristics of text during the neutralization process.
- The approach effectively handles longer input sequences and produce text that is both semantically coherent and fluent by using a transformer-based model as generator.
- We improve the training process and make the results more consistent, by incorporating an auxiliary guidance mechanism that guides the generator on which parts of the text are biased and need to be rephrased or removed. It also efficiently address multi-occurrence bias in a single sentence.
- We use a pre-trained cross-domain bias classifier as a self-supervised signal during adversarial training, eliminating the need for external labels. The cross-domain nature also helps to efficiently adapt to other domains, making it more practical for real-world applications.

Through seven models (3 baselines and 4 adversarial) along with human judgement evaluation, we demonstrate that the proposed approach outperforms conventional text style transfer techniques through a combination of subjective and objective evaluation metrics. Furthermore, both quantitative and qualitative evaluations on semantic similarity and fluency benchmarks demonstrate good preservation of semantic content of the original text, as well as coherent generated text.

## 2 Related Work

### 2.1 Subjective Bias Neutralization

Efforts to address bias neutralization in text often focus on demographic bias such as gender bias (Manzini et al., 2019; Zhao et al., 2017, 2019; Bordia and Bowman, 2019; Wang et al., 2018). Pryzant et. al (Pryzant et al., 2019) was the first to address generic linguistic bias correction using two types of models. Their modular model first detects bias and later performs the rewrite, whereas the concurrent model performs bias neutralization as an end-to-end task. Both methods achieve reasonable performance, but are restricted to single word bias (e.g., removing a single adjective). Additionally, both models resemble pointer

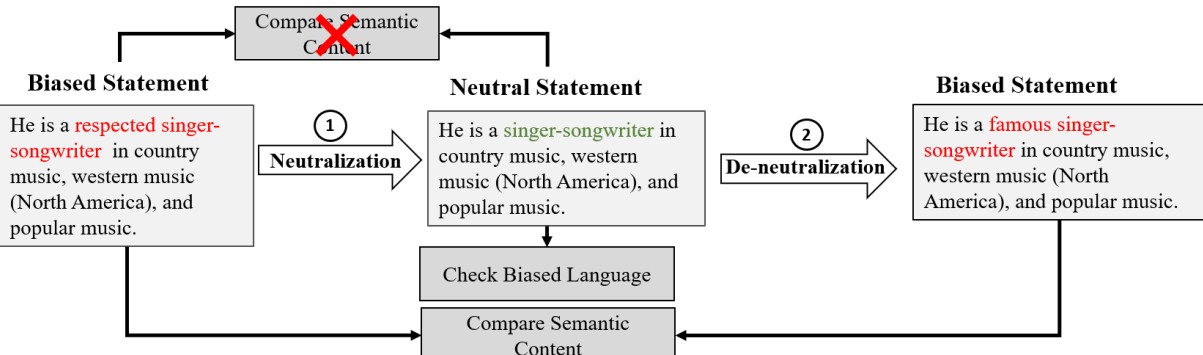

Figure 1: Overview of our proposed framework. Neutralization ensures the subjective bias is removed and the de-neutralization ensures the semantic content is preserved.

networks that attempt to maximise the occurrence of the same words in the source sentence, resulting in generated sentences that can lack fluency and diversity. Liu et al. (Liu et al., 2021) worked on depolarizing political text. In both these approaches, text segments (words or sentences) that are subjective or polarizing are first identified and then replaced with those that are semantically similar but less subjective. Madanagopal et al. (Madanagopal and Caverlee, 2023) proposed a reinforcement learning-based method to improve the performance of supervised seq2seq models. They also addressed the problem of multi-word bias. It is worth noting that all existing models for bias neutralization have been supervised models that rely on labeled datasets for single-word bias neutralization. In contrast, this paper targets self-supervised bias neutralization while also addressing multi-word bias, with an emphasis on fluency and diversity in the generated sentences.

## 2.2 Text Style Transfer

Bias neutralization shares many similarities with text style transfer. Typically in text style transfer, an input text is reconstructed so that linguistic attributes are transferred from one value to another while preserving content (Jin et al., 2022; Lample et al., 2018). Supervised methods on language style transfer typically use a sequence-to-sequence (seq2seq) neural machine translation model to transfer attribute styles using parallel corpora (Briakou et al., 2021; Madaan et al., 2020; Prabhumoye et al., 2018; Pryzant et al., 2019). In the absence of parallel corpus, text style transfer relies on disentanglement, prototype editing, and pseudo-parallel corpus construction (Jin et al., 2022). With disentanglement, the content and linguistic attributes of interest such as sentiment are separated using variants of autoencoders, and the disentangled content is rewritten with the target

style (Liu et al., 2020; Yi et al., 2021; Jin et al., 2020). However, these studies have observed that the overlap between the generated and ground-truth data is very low, indicating that the autoencoder model tends to discard too much information from the original text (Hu et al., 2017; Bowman et al., 2015). While these methods focus mainly on changing the attribute style of the text, they do not necessarily preserve the content. Our proposed method draws inspiration from attribute style transfer techniques. However, we specifically focus on addressing content-leaking that occurs during training for subjective bias neutralization using non-parallel data.

## 2.3 Generative Adversarial Networks

GANs have shown remarkable results in image style transfer with non-parallel data (Goodfellow et al., 2020). One limitation of using GANs in the text domain is the non-differentiable nature of discrete word tokens. Despite these challenges, GANs have been effectively applied to text style transfer (Yang et al., 2018; Liu et al., 2021). Liu et al. paired a transformer-based language model with a GAN network to neutralize and reverse the polarity of news articles (Liu et al., 2021), while Yang et al. employed a conditional GAN-based framework towards diverse paraphrase generation of text (Yang et al., 2018). Fedus et al. developed MaskGAN, integrating GAN with actor criticism to complete missing text based on the context (Fedus et al., 2018). Zhu et al. proposed CycleGAN, a GAN variant to learn image-to-image translation by leveraging two GAN networks in sequence to improve content preservation, functioning on the principle of cycle-consistent adversarial networks (Zhu et al., 2017). Wang et al. evaluated cycle training-based methods for generating text from structured data and showed the performance is on-par with supervised trained models (Wang et al.,

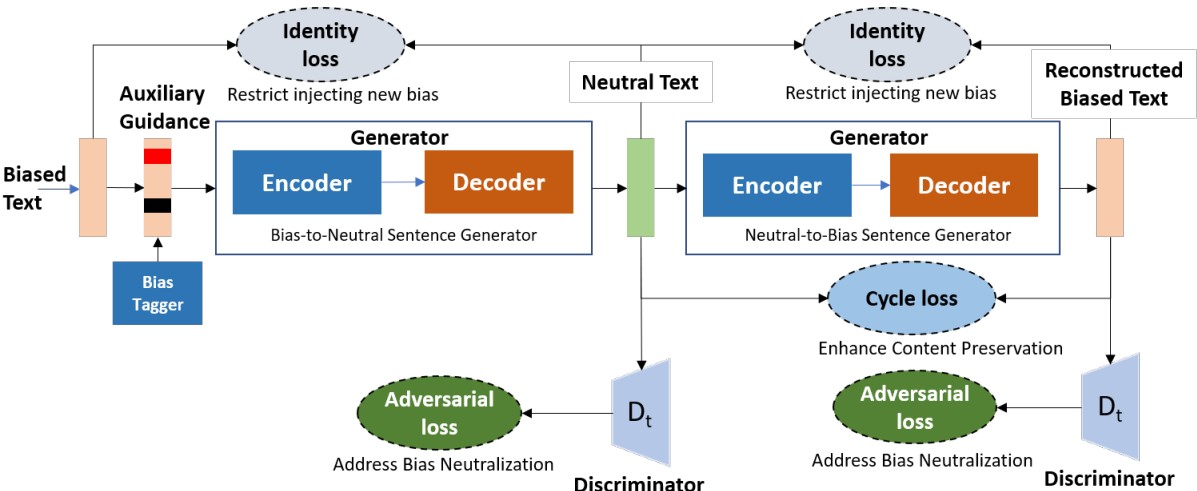

Figure 2: Illustration of the proposed cycle-consistent training based neutralization approach for improved content preservation. The cycle consistency loss ensures the preservation of bias-independent content, the adversarial loss works to neutralize bias, and the identity loss prevents over-correction that could introduce new bias.

2023). The core of our proposed approach is inspired by this cycle-consistent adversarial training approach. Our approach incorporates auxiliary guidance for better performance and reduced training time, as well as a novel prompt variation thus minimizing content loss and improving subjective bias neutralization accuracy.

## 3 Problem Statement

We assume we have a corpus $X$, composed of biased statements $x_1, x_2, ..., x_m$, and an additional corpus $Y$ containing neutral statements $y_1, y_2, ..., y_n$. These corpora exhibit non-parallelism, with no direct correspondence between their respective elements, such as $x_1$ and $y_1$. Our goal is to develop a model that corrects the subjective bias in the text and generates a neutral-tone output while preserving the bias-independent content of the original text. For simplicity, let's consider that $x$ represents an input biased text and $y$ represents a transformed neutral text. The subjective bias in text $x$ can be modeled as a function $f(x)$, which we aim to reduce or eliminate while generating the neutral text $y$.

For any given $x$, the model should generate $y$ such that the following conditions are satisfied:

- The bias in $y$ should be minimal. $f(y) \approx 0$

- The bias independent content from $x$ should be retained in $y$. If $g(x)$ represents a function that extracts the bias independent content from $x$, then we want $g(x) \approx g(y)$

- The generated text $y$ should be fluent and diverse. If $h(x)$ represents the fluency of the text $x$. then we want $h(y) > h(x)$

Therefore, the objective function is to minimize $f(y)$, subject to: $g(y) \approx g(x)$ and $h(y) > h(x)$. Hence, we need to establish mapping functions between these biased $X$ and neutral statements $Y$, and reciprocally, by only leveraging unpaired samples drawn from the distributions $p_d(x)$ and $p_d(y)$.

## 4 Method: FairBalance

In this section we present FairBalance, which enables the training of a subjective bias neutralization model in the absence of parallel data. The goal of FairBalance is to effectively transform biased text into a neutral form, showcasing both fluency and diversity. More importantly, it achieves this subjective bias transfer while preserving the underlying semantic content of the original biased text.

### 4.1 Overview

Given our primary objective of performing bias style transfer with non-parallel data (Section 3), GANs are a natural fit. However, they often suffer from limited and repetitive outputs, resulting in a lack of diversity in the generated text. This phenomenon of mode collapse occurs when the generator successfully deceives the discriminator to such extent that it fails to provide adequate feedback, resulting in a failure of training process. To mitigate mode collapse, we adopt a cyclic consistent generative adversarial approach for bias neutralization.

Such a cyclic approach consists of two sequential GAN networks $G_{XY} : X \rightarrow Y$ converts biased text to neutral text, and $G_{YX} : Y \rightarrow X$ performs the inverse conversion. Each generator is accom-

panied by a discriminator. The goal is to learn the mapping function between the source distribution X and target distribution Y that minimizes $f(y)$, while also preventing mode collapse. To address mode collapse and preserve bias independent content ($g(x) \approx g(y)$), such a cyclic approach introduces a cycle consistency loss that is computed by comparing the original input biased text with the reconstructed biased text (second network) (Zhu et al., 2017). Additionally, the text generated by both the generators are validated for its neutrality by a subjective bias classifier. Consequently, we can generate more diverse and realistic samples ($h(y) > h(x)$) without suffering from mode collapse.

## 4.2 Training Objectives

The two generators and two discriminators are trained simultaneously with the following set of losses:

**Adversarial Loss** is responsible for distinguishing real data from fake data generated by the generator. The adversarial loss for both networks can be expressed as:

$$L_{adv}(G, D_y, X) = \frac{1}{m} \sum (1 - D_y(G(x)))^2$$

$$L_{adv}(F, D_x, Y) = \frac{1}{m} \sum (1 - D_x(F(y)))^2$$

**Cycle Consistency Loss** enforces a mapping from the source domain to the target domain and back, so that the final text generated by the second GAN network is as close as possible to the original. The cycle consistency loss can be expressed as:

$$L_{cycle}(G, F, X, Y) = \frac{1}{m}[(F(G(x_i) - x_i)) - x_i) \\ + (G(F(y_i)) - y_i)]$$

**Identity Mapping Loss** helps to regularize the generator to get close to an identity mapping when the original input text is already in the neutral form (target domain). The identity mapping loss can be formulated as:

$$L_{ident}(G, F) = E_{y \sim p_{data}(y)}[\|G(y) - y\|_1] \\ + E_{x \sim p_{data}(x)}[\|F(x) - x\|_1]$$

By utilizing an adversarial loss, the generator generates neutral text that is validated by the discriminator and the cycle consistency loss ensures the bias independent content is preserved.

## 4.3 Auxiliary Guidance

Naively applying such a cyclic GAN architecture to bias neutralization can be challenging however. These methods, while sophisticated, can take a long time to converge. Further, the high dimensional output space and larger vocabulary make it computationally challenging to converge on large datasets, leading to elongated convergence times and, in certain instances, preventing models from reaching equilibrium entirely. Additionally, the sequential nature of text required the models to capture long-range dependencies between words, which further leads to model instability.

To address these issues, we introduce an auxiliary guidance as a mask that informs the generator what part of the sentence needs to be corrected. However, identifying what part of the text needs to be corrected is a challenging task. Concretely, we use the WikiBIAS dataset and train a bias tagger using a seq2seq model as detailed in (Zhong, 2021). This model identifies the section of text that is biased, based on the contextual usage, and highlights it for neutralization. This approach is able to identify multiple instances of bias within a single sentence, demonstrating strong performance with an accuracy of 95% with a recall of 92% on the validation set. This approach, leveraging auxiliary guidance, helps in maintaining control over the quality of neutralization and enhances the robustness and dependability of the bias-neutralization model.

## 4.4 Cross-domain Discriminator Learning

For GAN-based bias neutralization model to detect and correct subjective biases across multiple domains, the discriminator must be able to accurately identify many forms of bias and compute losses accordingly. Since our training data is derived solely from one domain (Wikipedia), training a discriminator with it can affect the model's performance and generalizability across other domains. Hence, we propose to pretrain the discriminator using a cross-domain bias classification dataset (Madanagopal and Caverlee, 2022). The cross-domain bias classification dataset contains a collection of subjectively-biased and neutral statements collected from various domains such as politics, entertainment and e-commerce. The cross-domain data helps the discriminator learn to identify patterns and features that are common across various domains. The use of an external classifier as discriminator helps to leverage its gained knowledge in the GAN and reduces the model

training time significantly.

## 4.5 Model Training

We use a transformer model for our generator network and pre-trained a subjective bias classifier for the discriminator. Since the bias classifier is pre-trained completely, the weights of the model are frozen when used as a discriminator and only the generator weights are updated during the training process. All three losses in the previous sections are important for the bias neutralization task. To produce a high-quality bias neutralization, all three losses need to be combined with a good balance before using it to update the generator algorithm. To make it flexible to test various configurations of the losses, the loss weights ($\lambda_{cycle}$ and $\lambda_{identity}$) are made configurable by the user.

$$
\begin{aligned}
L_{Generator} = {} & L_{adv}(G, D_y, X, Y) \\
& + L_{adv}(F, D_x, Y, X) \\
& + \lambda_{cycle} * L_{cycle}(G, F, X, Y) \\
& + \lambda_{ident} * L_{ident}(G, F)
\end{aligned}
$$

During the initial stages of the training process, we noticed that the generator was generating text with repeated words, which allowed the discriminator to easily beat the generator. In order to bootstrap our generator to generate reasonable text even from the beginning, we pretrained it as an autoencoder where the generator is expect to produce an output that is similar to the input text. Both the forward and inverse generators were trained using a similar approach.

Also, in order to prevent the model from changing drastically from iteration to iteration, the discriminators were fed a history of generated text, rather than just the ones produced by the latest versions of the generator. To do this we keep storing the 50 most recently generated text. Based on this technique we reduce the model oscillation as well as model overfitting.

## 5 Experiment Setup

This section provides an overview of the datasets, the baseline bias neutralization models, and the detailed evaluation results.

## 5.1 Datasets

For training the discriminator, we used a cross-domain bias dataset that was constructed from diverse sources, including Wikipedia, and other subjectivity rich domains such as MPQA Opinion corpus (Wiebe et al., 2005) and Ideological

Book Corpus (IBC) (Sim et al., 2013). A total of 412,188 sentences was curated for training cross-domain bias classifier.

The non-parallel dataset for GAN training was derived by analyzing the edit histories of Wikipedia articles that were tagged for Neutral-point-of-View (NPOV) issues. We followed the data harvesting approach similar to (Pryzant et al., 2019; Recasens et al., 2013) with modifications to support multi-word, multi-occurrence bias. We had a total of 201,175 biased sentences and 296,504 neutral sentences for building subjective bias neutralization model. More details of the data preparation steps are available in Appendix A.

## 5.2 Baseline Models

In the following, we present the baseline models that are developed to evaluate the performance of the proposed auxiliary guided, cycle-consistent training-based subjective bias neutralization method.

**Delete Biased Word(s)** uses a biased word lexicon that is curated from various studies to identify biased word in the input sentence and removes them based on certain linguistic rules.

**Control-gen Model** is a neural text generation model developed by (Hu et al., 2017) that uses a variational autoencoder as the generator and a discriminator for conditioning the generated text.

We also explored a diverse set of GAN-based models aiming to gain insights into their respective approaches for subjective bias neutralization:

**GAN**: This model employs a transformer-based generator along with a pre-trained bias classifier as the discriminator. It serves as a fundamental GAN architecture for our experiments.

**MaskGAN**: The MaskGAN-based model incorporates a bias tagger to identify and mask word tokens that are considered subjectivity indicators (Fedus et al., 2018). Similar to the GAN model, it utilizes a transformer-based generator and a pre-trained bias classifier as the discriminator.

**CycleGAN**: Designed as a cycle-consistent adversarial network, the CycleGAN-based model adopts transformer-based architectures for both generators (Yang et al., 2018). It employs the same pre-trained classifier with an inverse probability score as the discriminator.

## 5.3 Experimental Results

In this section, we present the results of the experiments through both automated and human judge-

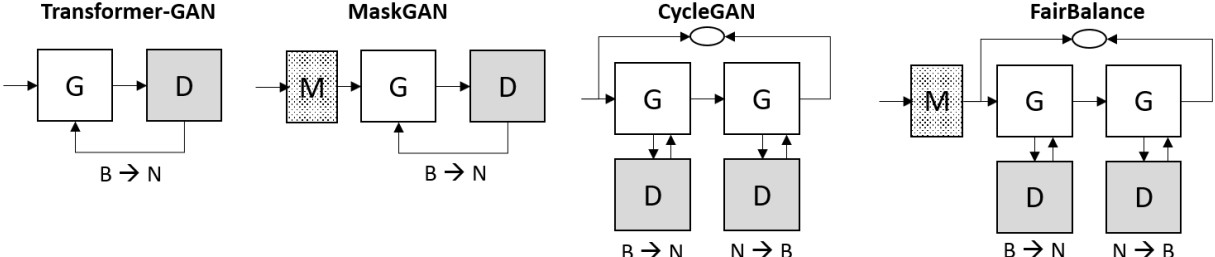

Figure 3: Illustration of the various GAN-based models experimented to gain insight into their respective approaches for bias neutralization. $B \rightarrow N$ represents converting biased text to neutral and $N \rightarrow B$ represents converting neutral text to biased. M represents the masking that highlights biased spans in the input sentence.

| Model | Neutrality↑ | BLEU↑ | BERT-Score↑ | PPL↓ |
|---|---|---|---|---|
| Source Copy | 12.88 | 100.00 | 100.00 | 35.14 |
| Delete Biased Word | 39.45 | 79.71 | 92.23 | 48.20 |
| Control-gen (Hu et al., 2017) | 69.18 | 41.64 | 78.31 | 29.47 |
| GAN | 59.85 | 30.81* | 91.58* | 27.17* |
| MaskGAN | 67.26* | 23.61* | 93.36* | 27.92* |
| CycleGAN | 66.94* | 40.30* | 95.65* | 25.61* |
| FairBalance | **69.78*** | **51.38*** | **96.42*** | **26.48*** |

Table 2: Performance comparison Subjective Bias Neutralization models using various rule-based and neural GAN-based style transfer models. For quantitative metrics, rows with asterisks (*) are significantly different than the preceding row. ↑/↓ means higher/lower score is preferred for the corresponding metric.

ment evaluation. The details of evaluation metrics used are available in Appendix E.

### 5.3.1 Automated Evaluation

Table 2 shows the performance of our proposed model and its variants, in comparison with the selected baseline models. Initially, we evaluated the performance of our non-GAN based models (baselines) to comprehend the strengths and weaknesses of our baseline models. Since more than 30% of the biased words in our corpus were adjectives, the method of deleting biased words proved more effective by eliminating the adjectives and neutralizing the sentences. However, this approach resulted in grammatical issues, such as incorrect usage of determiners and articles.

Alternately, the neural network based control-gen model developed by Hu et al. (Hu et al., 2017) neutralized the sentences more effectively than the delete biased word method. This can be mainly attributed to the discriminator used to control the generation process. Despite the control-gen method's superior neutralization performance and the fluency of the output text, there was a significant loss of content. Sometimes, it also added additional content to the neutralized text. While the generated text retained the overall theme of the input, some important facts were omitted.

Next, we analyze the neutralization perfor-

mance and training performance of the GAN-based models. The vanilla GAN-based model, which employs a transformer model as its generator and a simple RoBERTa based discriminator took a long time to train (weeks) and the generated text contained lots of repeated words. With the introduction of our pre-training approach, the transformer-GAN model started producing coherent text. For simple framing biases, such as use of weasel words, the transformer-GAN model was able to identify the biased terms in a sentence and neutralize it by removing it or replacing it with other words. However, it struggled to neutralize sentences that with quotes and uncertainties (epistemological bias). The masked GAN-based model, which marks the biased words in the input sentence, allowed for faster training (days) and achieved a neutrality score of 67.26 which is 8 points better than vanilla-GAN models. When sentence length exceeded 40 words, the masked transformer model began to lose significant original content from the input text.

Both the CycleGAN and FairBalance models aimed to balance neutralizing the input text with retaining portions of the original semantic content. Both models were able to neutralize long sentences and generate more fluent and grammatically correct sentences. The FairBalance model

had a 2 point improvement in neutrality score relative to other models and also it took less time it to converge. It was also capable of addressing multiple instances of bias within a single sentence.

Interestingly, the FairBalance-based models did not introduce new bias into the input text, whether it was originally neutral or biased. This aspect of neutralization is significant and the combination of identity loss and discriminator loss introduced in the FairBalance along with auxiliary guidance is responsible for this important neutralization aspect in addition to its improved content preservation.

### 5.3.2 Multi-Occurence Bias Evaluation

When it comes to addressing multi-occurrence bias, models incorporating auxiliary guidance or masking (MaskGAN and FairBalance) showed superior performance. The masking done with the token-wise bias tagger aids in identifying biased sections of text and instructs the generator to address them. Other GAN models such as the GAN and CycleGAN with a transformer, primarily addressed single occurrence of bias and retained other subtly biased chunks. Even though both models with auxiliary guidance performed well, the CycleGAN model with auxiliary guidance retained the bias independent content in the generated text better than the MaskGAN.

| Model | Neutrality | % biased |
|-----------|------------|----------|
| GAN | 61.19 | 30.57 |
| MaskGAN | 69.26 | 23.94 |
| CycleGAN | 64.09 | 29.54 |
| FairBalance | **70.48** | **22.16** |

Table 3: Performance evaluation of addressing multi-occurrence bias. % biased represents the percentage of sentences that contains at least one instance of biased chunk after bias neutralization (lower the better).

### 5.3.3 Human Evaluation

In the presence of parallel data, it is relatively easy to assess the performance of neutralization models using automatic metrics such as BLEU and BERTScore. In the absence of non-parallel data, we chose to evaluate the performance of our neutralization models through human evaluations to ensure unbiased feedback. Therefore, a blind heads-up comparison was used to evaluate the quality of samples generated by various GAN-based bias neutralization models. Three aspects of text were evaluated using human judgement: bias, fluency and content preservation. For every judge, we presented the original text and neutralized text and asked them to rate the quality of neutralized text in the scale of -2 to 2. In terms of bias,

the MaskGAN with Transformer model had a best bias score of -0.894 (See Table 4). Interestingly the masked transformer GAN model had a second best of 0.664. This shows the masking approach works better than other methods in making the text neutral. In terms of fluency, most of the models were in the same range of 0.1-0.12. FairBalance had the best fluency score of 0.115. In terms of content preservation, text generated by Cycle-GAN based models were preferred by the users with the FairBalance with transformer model having the best content preservation score of 1.394. Overall, both masking based models were scored high in the human judgement. Given both neutralization and content preservation are important, FairBalance is the best performing model.

### 5.4 Domain adaptation

To evaluate the domain adaptation of the proposed FairBalance model on domains outside of Wikipedia, we collected 25 statements from each of the following domains: Academics, News, and Politics. This resulted in a total of 75 sentences, which were then presented to five human judges for qualitative assessment. Three criteria were evaluated: neutrality, content preservation, and fluency. In terms of neutrality, the FairBalance model exhibited robust performance across all domains, outperforming the second-best model, MaskGAN (See Table 5). The FairBalance model demonstrated particularly strong performance in the Academics domain. With respect to fluency, all models performed reasonably well. Excluding the Academics domain, FairBalance showed strong performance across the remaining domains. As for content preservation, there was a marked improvement in the performance of FairBalance relative to the other models. The human evaluation revealed that the FairBalance model outperformed other models across different domains.

## 6 Conclusion

In this paper, we propose a new subjective bias neutralization model trained using non-parallel data through an auxiliary guided cycle consistent GAN. FairBalance consists of a bias tagger to identify subjective words in the text, a cross-domain bias classifier that accurately detects whether the given text is subjectively biased or not and a cyclic network that trains the neutralization model with a combination of adversarial, cycle-consistency and identity mapping loss. The combination of auxiliary guidance, pre-trained clas-

| Model | Bias↓ | Fluency↑ | Content↓ |
|---|---|---|---|
| GAN | -0.066 | 0.108 | 2.165 |
| MaskGAN | -0.664 | 0.106 | 1.898 |
| CycleGAN | -0.648 | 0.112 | 1.501 |
| **FairBalance** | **-0.894** | **0.115** | **1.394** |

Table 4: Human Evaluation of Subjective Bias Neutralization models using four different GAN-based text style transfer models. The rating scale for bias ranges from -2 to 2, fluency ranges from -2 to 2, and content preservation ranges from 0 to 4. ↑ /↓ means higher/lower score is preferred for the corresponding metric.

| Model | Academics | | | News | | | Politics | | |
|---|---|---|---|---|---|---|---|---|---|
| | Bias↓ | Fluency↑ | Content↓ | Bias↓ | Fluency↑ | Content↓ | Bias↓ | Fluency↑ | Content↓ |
| GAN | -0.617 | 0.085 | 2.97 | -0.489 | 0.105 | 2.17 | -0.427 | 0.092 | 2.03 |
| MaskGAN | -0.826 | 0.089 | 2.25 | -0.715 | 0.110 | 2.09 | -0.524 | 0.102 | 2.48 |
| CycleGAN | -0.828 | **0.103** | 1.45 | -0.684 | 0.109 | 1.58 | -0.503 | 0.097 | 1.16 |
| FairBalance | **-0.901** | 0.102 | **0.93** | **-0.746** | **0.112** | **1.53** | **-0.584** | **0.105** | **1.04** |

Table 5: Human evaluation of subjective bias neutralization performance across 3 different domains. The rating scale for bias ranges from -2 to 2, fluency ranges from -2 to 2, and content preservation ranges from 0 to 4. ↑ /↓ means higher/lower score is preferred for the corresponding metric.

sifier and cycle-consistent network yielded significant improvement in performance when comparison with other GAN-based models. Further, the FairBalance model also performed well on other domains which is evident through the cross-domain bias evaluation.

While the proposed model performs well, there is still scope for improvement. For example, incorporating more contextual information such as paragraphs or preceding sentences can improve the detection and neutralization performance. Additionally, the performance of FairBalance depends on the bias tagger performance. By incorporating more subtle forms of subjective biases in refining the bias tagger improved the performance of proposed method. In future studies, we plan to extend our proposed framework model to work on long sentences and more subtle forms of biases. We will also explore paragraph level bias neutralization to make the generated text more consistent and accurate by using more contextual data.

## 7 Limitations

FairBalance shares certain limitations with previous research (Kim et al., 2017; Yi et al., 2017) on cyclic adversarial-based learning methods. One of the major limitations is that the mappings learned are deterministic and the cycle consistency forces the mappings to be inverses of each other (Alma-hairi et al., 2018). Instead of capturing true, structured conditional distribution, these cyclic adversarial-based models arbitrarily pick a one-to-one mapping in situations when faced with complex source-target relationships.

Additionally, this study focuses on a specific forms of subjective bias observed at the sentence level. However, considering the context (paragraph) in which a statement is made could potentially alter the perspective. It is important to note that while the FairBalance approach is evaluated on a real-world subjective bias neutralization task in this study, further testing is necessary to encompass more challenging bias types and explore other target model architectures.

The work presented in this paper introduces a promising approach for developing bias neutralization models in domains like news and politics where parallel data is not available, which is a field of great significance yet remains under explored. We hope that by evaluating these models across different domains, it will stimulate further research in developing robust, nuanced, and fair bias neutralization models.

## 8 Ethics Statement

In our exploration of machine learning models for neutralizing linguistic bias, we acknowledge and emphasize the inherent limitations and challenges. Specifically, our FairBalance model bases its analysis on individual sentences without accounting for the broader context in which they appear. This absence of context might lead to instances where the model misidentifies a sentence as either neutral or biased. We recognize that this potential inaccuracy could influence its operational utility. Readers and practitioners should be cautious and considerate of this limitation when interpreting or deploying the results from our model.

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

## A Dataset Preparation

Our data harvesting approach was similar to Recasens et. al (Recasens et al., 2013) and Pryzant et. al (Pryzant et al., 2019), but with some minor changes such as not restricting to single word edits. Sentences that had NPOV tags before the revision were considered as biased sentences and the sentences whose NPOV tags were removed after edits were considered as unbiased sentences.

We ignored revisions that were related to missing references, misspellings and punctuation. Additionally, we downloaded Wikipedia pages that are tagged as "Good Articles" and included them in the neutral sentences corpus (Wik, 2022).

Since the objective of this research is to correct bias that is induced by single word and multiword, we expanded the corpus by modifying the harvest function of pryzant et. al. (Pryzant et al., 2019) Also, our method uses the latest dump from Wikipedia which contains new biased sentences that were not considered in the previous study. Additionally, some data cleanups were done to make this model not sensitive to proper nouns in the text. We replaced all proper nouns with generic names, but retained honorifics because some of the gender biases were introduced through honorifics. The numbers mentioned in the text were also replaced with a NUM tag.

## B  Baseline Model Selection

Our study aimed to comprehensively evaluate both disentanglement-based and adversarial-based models in the context of our bias neutralization dataset. This approach was motivated by the need to explore the strengths and weaknesses of different techniques in addressing bias neutralization using non-parallel corpus. In the realm of disentanglement-based models, our investigation led us to the discovery that the control-gen method (Hu et al., 2017) exhibited the most promising performance for bias neutralization within our bias neutralization dataset. This result came as a surprise, particularly given the prevailing expectation of better performance from more recent models (Yi et al., 2021; Jin et al., 2020). That's why we used only the control-gen method for deeper analysis on disentanglement methods. Since our method was adversarial-based, we choose one disentanglement method (best) and more than one adversarial model for our evaluation.

For adversarial-based models, we recognized that while CycleGAN (Yang et al., 2018) and MaskGAN (Fedus et al., 2018) were relatively older references, we did not simply adopt these models as-is for our study. Instead, we took steps to enhance their relevance and applicability. We incorporated state-of-the-art transformer models to leverage advancements in natural language processing, ensuring that our adversarial-based models were equipped with the latest language understanding capabilities. We used RoBERTa models for training the pre-trained classifier for dis-

criminator. Moreover, we employed a bias tagger for the purpose of masking, which further aligned these models with our dataset's specific requirements. By adapting these models with modern components and techniques, we aimed to bridge the gap between their original designs and the demands of our bias neutralization task. This approach enabled us to harness the advantages of both established and cutting-edge methodologies, ensuring that our evaluation was as comprehensive and effective as possible.

## C  Implementation Details

All the GAN models were implemented in PyTorch (Python 3.7.11) and trained using a Tesla V100 with CUDA 11.3. For discriminator training, a pre-trained contextualized language models is used to efficiently incorporate sentence semantics in performing text classifications (Madanagopal and Caverlee, 2022). The pre-trained language model, RoBERTa was downloaded and further trained using cross-domain dataset (Liu et al., 2019). We used ADAM optimizer with a learning rate of 2e-4 and the trained discriminator had a precision of 89% with an F1 Score of 87%. The input sentence $s$ is first encoded into a fixed length vector $h$ using pre-trained RoBERTa model, which captures the contextual information of the sentence. Using a fully connected layer with softmax function, the encoded vector is transformed to a probability distribution over the possible labels which is given by:

$$p(y|s) = softmax(W[h + b]) \qquad (1)$$

where W is the weight matrix of the fully connected layer and b is the bias vector.

| Model | Precision | Recall | F1-score |
|---|---|---|---|
| CLS$_{GloVe}$ | 81.81 | 80.24 | 81.02 |
| CLS$_{BERT}$ | 83.57 | 79.26 | 81.36 |
| CLS$_{RoBERTa}$ | **89.41** | 85.94 | **87.64** |

Table 6: The CLS$_{RoBERTa}$ trained with cross-domain dataset showed significant improvement in classifying biased statement. **Bold** indicates best results.

The generator model used in all four GAN models consists of three transformer layers with three attention heads. In order to capture the underlying semantics of the input text, we pretrained the generators as autoencoders. The maximum length of the text to be generated was set to 128. To opti-

mize the networks, we employed the Adam optimizer with a learning rate of 0.0002, betas of (0.9, 0.98), and an epsilon value of 1e-9. To improve the stability and convergence, we computed gradient penalty using gradient norm with a weight $\lambda = 10$ and added it to the discriminator loss during the GAN training. For the encoder and decoder, a dropout rate of 0.33 was applied. During the training of the generator, we utilized 25 epochs, while for the FairBalance training, we used 100 epochs. Each training iteration was performed with a batch size of 16. To assign appropriate importance to different components of the loss function, we set the loss weights as $\lambda_{cycle} = 8$ and $\lambda_{identity} = 5$. These weights help in balancing the impact of cycle consistency and identity preservation during the training process.

## D Bias Tagger

The bias tagger is a token classification model used to analyze input sentences and identify biased spans of text. It is developed in PyTorch using RoBERTa for the token classification transformer model. The Adam optimizer with a learning rate of 3e-5 was used. The model is trained using the WikiBIAS dataset developed by Zhong et al. [cite: zhong2021wikibias]. The trained bias tagger model achieved 95% accuracy in identifying biased spans of text with a recall of 92%. In the FairBalance model, the input sentence is first fed to the bias tagger to identify biased spans of text. This approach helps identify multi-word biased spans as well as multiple occurrences of biased text. The output of the bias tagger is then used to add $[MASK]$ tokens around biased words, which are subsequently fed as input to the first GANs generator network.

## E Evaluation metrics

To assess the performance of our proposed bias neutralization model, we used the following automatic evaluation metrics.
**Neutrality:** Similar to (Luo et al., 2019; He et al., 2020), We used a subjective bias classifier to compute the neutrality score for an input sentence. The subjective bias classifier is a BERT-based model that was trained using subjective statements derived various domains such as political speeches and product reviews (Madanagopal and Caverlee, 2022). On a human reference dataset, this bias classifier had an accuracy of 89% (Madanagopal and Caverlee, 2022)

**BLEU:** A set of 100 biased sentences were selected for this evaluation and corresponding neutral sentences were manually created with human experts who have good English language skills. Using this reference dataset, n-gram precision counting metric (BLEU) was computed to measure the similarity of machine corrected text and human reference correction through n-gram precision counting (Papineni et al., 2002).
**BERTScore:** Uses contextual language models and computes semantic distance between candidate and reference sentences (Zhang* et al., 2020) using a contextualized language model like BERT.
**PPL:** To evaluate the grammatical correctness and fluency of the machine generated text, we computed the perplexity score (PPL) using the large pre-trained language model GPT-2. The perplexity score PPL is computed directly on the generated text with no reference text.

## F Human Judgement

A set of 50 biased sentences were selected and processed using the experimented GAN models. This combination of biased and neutral sentences was used for human evaluation by a panel of 10 judges. Every time a biased sentence and its corresponding neutralized sentence from one of the model is presented to the user and asked to rate the following three aspects separately:

- **Neutrality**: Rate the neutrality of the neutralized sentence on a scale of -2 to 2. A rating of -2 indicates that the original text is more biased than the neutralized text, a rating of 0 means both texts have the same level of bias, and a rating of 2 means the neutralized text is more biased.
- **Content Preservation**: Rate whether the bias independent content is retained in the neutralized sentence on a scale of 0 to 4 with 0 means the content is very different and 4 means very similar.
- **Fluency**: Rate the grammatical correctness and fluency of the neutralized sentence with respect to the original text on a scale of -2 to 2 with -2 mean the original text is more fluent and 2 means the neutralized text is more fluent.

| Examples of Bias Neutralizations | |
|---|---|
| Source | *Unfortunately*, Terry Fox could not finish his *courageous* run. |
| GAN | *Unfortunately*, Terry fox could not finish. |
| MaskGAN | *Sadly*, Terry fox could not finish. |
| CycleGAN | *Courageous* Terry Fox could not finish his run. |
| FairBalance | Terry fox couldn't finish his run. |
| Source | Critics believe that some or all of these *exclusionary* policies are wrong and amount to discrimination. |
| GAN | Critics think that some or all of the policies are wrong. |
| MaskGAN | Some critics believe that these *discriminatory* policies are wrong. |
| CycleGAN | Critics believe that some of the policies are wrong and could lead to discrimination. |
| FairBalance | According to critics, some or all of these policies are wrong and discriminatory. |
| Source | Carleton University is a *prestigious* non-denominational, co-educational university in Ottawa, Ontario, Canada. |
| GAN | Carleton University is a public university. |
| MaskGAN | Carleton University, Canada is a non-denominational, co-educational university. |
| CycleGAN | Carleton University is a non-denominational, co-educational university in Ottawa, Ontario, Canada. |
| FairBalance | The Carleton University is a non-denominational, co-educational university located in Ottawa, Ontario, Canada. |
| Source | Joe is *respected* singer-songwriter in country music, western music (North America), and popular music. |
| GAN | Joe sings in country music, western music, and popular music. |
| MaskGAN | Singer Joe specializes in country music, western music (North America), and popular music. |
| CycleGAN | Joe is a singer-songwriter in country music, western music (North America), and popular music. |
| FairBalance | Joe is regarded as a leading singer-songwriter in country music, western music (North America), and popular music. |
| Source | Redstone's son has sued him and his daughter regarding the managing of National Amusements. |
| GAN | *Surprisingly*, Redstone's son sued him and his daughter for managing National Amusements. |
| MaskGAN | Redstone was sued by his son and daughter over the handling of National Amusements. |
| CycleGAN | Redstone's son has sued him and his daughter over managing National Amusements. |
| FairBalance | A lawsuit has been filed against Redstone and his daughter by their son over the management of National Amusements. |

Table 7: Example outputs from GAN-based neutralization models. Biased words are highlighted in *italic*.