# OpenReview forum: "Bias Neutralization in Non-Parallel Texts: A Cyclic Approach with Auxiliary Guidance"
_EMNLP/2023/Conference — EMNLP 2023 Main_

### Official Review · Reviewer_fXMK · 2023-08-04

**Soundness:** 3

**Excitement:**

3: Ambivalent: It has merits (e.g., it reports state-of-the-art results, the idea is nice), but there are key weaknesses (e.g., it describes incremental work), and it can significantly benefit from another round of revision. However, I won't object to accepting it if my co-reviewers champion it.

**Missing References:**

In section 5.2 (description of baseline models), there were no citations and references for the baseline models, MaskGAN and CycleGAN. I noted that references have been made in related work but it should also be re-referenced in the main section that describe the baseline models.

**Paper Topic And Main Contributions:**

This paper is on subjective bias neutralization on text without the use of parallel text corpus. The paper first discuss the 3 limitation of existing works on subjective bias neutralization, namely 1) the reliance of parallel text corpus, where the resource is limited and expensive, 2) the limited adaptation capability of domains from existing works, and 3) the challenge of preserving the content while removing the bias. The proposed approach is on using a cyclic adversarial network to handle both subjective bias neutralization and content preservation. The experimental results with baseline models show that the approach is promising.

**Reasons To Accept:**

The work is interesting and the proposed approach aims to tackle several limitation of existing works.

**Reasons To Reject:**

I have a main concern on the baseline models used. All the baseline models are kind of old. MaskedGAN and CycleGAN, the two main baseline models used in this work, are introduced in 2018 and 2017, respectively. Are there no recent models that can be used to compare? In the related work sections, there were multiple recent works that were described - the proposed approach should compare with them, or at least some of them. Without comparison with more recent models, the evaluation seemed to be unfair.

**Reproducibility:**

3: Could reproduce the results with some difficulty. The settings of parameters are underspecified or subjectively determined; the training/evaluation data are not widely available.

**Reviewer Confidence:**

4: Quite sure. I tried to check the important points carefully. It's unlikely, though conceivable, that I missed something that should affect my ratings.

---

> ### Author Rebuttal · Authors · 2023-08-28
>
> **I have a main concern on the baseline models used. All the baseline models are kind of old. MaskedGAN and CycleGAN, the two main baseline models used in this work, are introduced in 2018 and 2017, respectively. Are there no recent models that can be used to compare? In the related work sections, there were multiple recent works that were described - the proposed approach should compare with them, or at least some of them. Without comparison with more recent models, the evaluation seemed to be unfair.**
>
> We appreciate your perspective on the selection of models that were a little outdated and would like to elaborate on the justifications for our approach based on the findings from our study. Our study aimed to comprehensively evaluate both disentanglement-based and adversarial-based models in the context of our bias neutralization dataset. This approach was motivated by the need to explore the strengths and weaknesses of different techniques in addressing bias neutralization using non-parallel corpus.
>
> In the realm of disentanglement-based models, our investigation led us to the discovery that the control-gen method exhibited the most promising performance for bias neutralization within our bias neutralization dataset. This result came as a surprise, particularly given the prevailing expectation of better performance from more recent models. That’s why we used only the control-gen method for deeper analysis. Since our method was adversarial based, we choose one disentanglement method (best) and more than one adversarial model for our evaluation.
>
> Coming to the adversarial-based models, we recognized that while cycleGAN and MaskGAN were relatively older references, we did not simply adopt these models as-is for our study. Instead, we took steps to enhance their relevance and applicability. We incorporated state-of-the-art transformer models to leverage advancements in natural language processing, ensuring that our adversarial-based models were equipped with the latest language understanding capabilities. We used RoBERTa models for training the pre-trained classifier for discriminator. Moreover, we employed a bias tagger for the purpose of masking, which further aligned these models with our dataset's specific requirements.
>
> By adapting these models with modern components and techniques, we aimed to bridge the gap between their original designs and the demands of our bias neutralization task. This approach enabled us to harness the advantages of both established and cutting-edge methodologies, ensuring that our evaluation was as comprehensive and effective as possible.
>
> We are committed to transparency in our approach and have taken your feedback to heart. To ensure clarity in our paper, we will include a section in the methodology that explicitly discusses our choice of baseline models and our reasons for not comparing with recent disentanglement models. By doing so, we hope to provide a clear explanation for our approach and to address any concerns related to the choice of baselines. A new appendix section called Baseline Selection will be added to the revised version of our paper.
>
> We already made changes to the paper, but open review doesn't allow us to upload the revised version.
> _____________________________
>
> **Missing References:
> In section 5.2 (description of baseline models), there were no citations and references for the baseline models, MaskGAN and CycleGAN. I noted that references have been made in related work but it should also be re-referenced in the main section that describes the baseline models.**
>
> Thank you for highlighting our oversight in not including references to MaskGAN and CycleGAN in the baseline models section. We've now added the references as you suggested.

---

### Official Review · Reviewer_htDX · 2023-08-05

**Soundness:** 3

**Excitement:**

4: Strong: This paper deepens the understanding of some phenomenon or lowers the barriers to an existing research direction.

**Paper Topic And Main Contributions:**

This paper is about bias neutralization in non-parallel texts. It proposes a new approach called FairBalance that uses a cycle consistent adversarial network to enable bias neutralization without the need for parallel text. The model design preserves bias-independent content and through auxiliary guidance, the model highlights sequences of bias-inducing words, yielding strong results in terms of bias neutralization quality. The paper also discusses the importance of avoiding bias in many heavily relied upon resources such as Wikipedia, scholarly articles, news sources, and emerging large language models.

FairBalance is a new approach proposed in this paper for bias neutralization in non-parallel texts. It has three unique features:

1. A cycle consistent adversarial network enables bias neutralization without the need for parallel text.
2. The model design preserves bias-independent content.
3. Through auxiliary guidance, the model highlights sequences of bias-inducing words, yielding strong results in terms of bias neutralization quality.

FairBalance uses a transformer-based model as a generator to produce text that is both semantically coherent and fluent. It also incorporates an auxiliary guidance mechanism that guides the generator on which parts of the text are biased and need to be rephrased or removed. This approach improves the training process and makes the results more consistent.

The paper also discusses the limitations of FairBalance and suggests future directions for improvement, such as incorporating more contextual information and refining the bias tagger by incorporating more subtle forms of subjective biases.

**Reasons To Accept:**

1. FairBalance can be trained effectively in the absence of parallel data, which is a significant advantage since parallel data is often difficult to obtain.
2. The model design preserves bias-independent content, which is important for maintaining the original meaning of the text.
3. FairBalance produces text that is both semantically coherent and fluent, which is important for readability and comprehension.
4. The auxiliary guidance mechanism incorporated in FairBalance helps to improve the training process and makes the results more consistent and reliable.
5. FairBalance yields strong results in terms of bias neutralization quality, as demonstrated by extensive experiments.

**Reasons To Reject:**

1. FairBalance's performance depends on the quality of the bias tagger used to identify biased language. If the bias tagger is not accurate, the performance of FairBalance will be affected.
2. The experiments conducted in the paper are limited to a specific dataset and domain, and it is unclear how well FairBalance would perform on other datasets and domains.
3. The paper does not provide a detailed analysis of the computational complexity of the proposed approach, which could be a concern for large-scale applications.


**Reproducibility:**

3: Could reproduce the results with some difficulty. The settings of parameters are underspecified or subjectively determined; the training/evaluation data are not widely available.

**Reviewer Confidence:**

2: Willing to defend my evaluation, but it is fairly likely that I missed some details, didn't understand some central points, or can't be sure about the novelty of the work.

---

> ### Author Rebuttal · Authors · 2023-08-28
>
> **FairBalance's performance depends on the quality of the bias tagger    used to identify biased language. If the bias tagger is not accurate,    the performance of FairBalance will be affected.**
>
> The key factor contributing to the enhanced performance of the FairBalance model is the integration of the bias tagger. In the absence of the auxiliary signal provided by the bias tagger, the neutrality performance of the CycleGAN model was 66.94. However, with the inclusion of the bias tagger, we achieved an improved performance of 69.78. It's noteworthy that training a bias tagger is a relatively straightforward process, and the FairBalance model's adaptability to new domains is facilitated by leveraging this component. A significant advantage of employing the bias tagger lies in the control it affords over the bias mitigation process. Unlike the less controllable black box nature of adversarial learning, which doesn't offer precise control over specific linguistic aspects to address, the bias tagger provides a more manageable and controllable approach. This controllability empowers us to precisely pinpoint and address the aspects of language that require mitigation, contributing to the model's effectiveness in bias reduction.
>
> _________________________________
>
> **The experiments conducted in the paper are limited to a specific dataset and domain, and it is unclear how well FairBalance would perform on other datasets and domains.**
>
> Even though our proposed FairBalance model was trained on Wikipedia data, we have performed experiments on its domain adaptability on other domains such as academics, news and politics and provided the results in Table 5.
>
> _________________________________
>
> **The paper does not provide a detailed analysis of the computational complexity of the proposed approach, which could be a concern for large-scale applications.**
>
> We value your feedback about the computational complexity of our proposed approach. While our primary focus in this paper was on bias neutralization using a non-parallel corpus, we noted the training durations of our models.
> For context:
>  - The RoBERTa-based discriminator training required approximately 3 hours on a V100 GPU with a batch size of 32.
>  - The sequence2sequence model for bias tagger finished training in around 8 hours under the same GPU and batch size conditions.
>
> All the adversarial models were trained for 200 epochs with a batch size of 16 on V100 with 16GB shared memory and following are the training times for the adversarial models:
>  - CycleGAN - 18 days 8 hours.
>  - Vanilla GAN model - 15 days 20 hours
>  - MaskGAN - 9 days 6 hours
>  - FairBalance model - 13 days 2 hours.
>
> Though we did not have details into theoretical computational complexities, these real-world training times should offer a practical perspective on the time efficiency of our models. We recognize the importance of this time complexity for large-scale applications, and will consider a more detailed analysis in our future works.

---

### Official Review · Reviewer_THyJ · 2023-08-09

**Soundness:** 3

**Excitement:**

3: Ambivalent: It has merits (e.g., it reports state-of-the-art results, the idea is nice), but there are key weaknesses (e.g., it describes incremental work), and it can significantly benefit from another round of revision. However, I won't object to accepting it if my co-reviewers champion it.

**Paper Topic And Main Contributions:**

The paper addresses the issue of subjective bias in text, which can manipulate our perception of reality and intensify social conflicts. It highlights the importance of avoiding bias in resources like Wikipedia, scholarly articles, and news sources. Existing methods for bias neutralization rely on parallel text and struggle with domain transfer and content preservation. Mainly, the paper introduces a new approach called FairBalance, which aims to expand the reach of bias neutralization. It utilizes a cycle-consistent adversarial network to neutralize bias without needing parallel text. The model design focuses on preserving bias-independent content and auxiliary guidance.

**Questions For The Authors:**

1. There is no information about human judges, even though that might be very relevant. How was their education, age, etc.? This information is increasingly included in recent works about bias and relates to their labels.

2. The study focuses on specific forms of subjective bias observed at the sentence level. However, considering the context (paragraph) in which a statement is made could potentially alter the perspective. You say one sentence on this, but do you already have ideas on how it will be implemented in your work?

3. The referenced bias-neutralizing papers are not always completely clear in which kind of bias they deal with. However, bias (from linguistic bias to sentiment bias and bias by word choice and other concepts) is very complex. Can you clarify the definition of bias that you apply?

4. There seems to be no link to all data and code (anonymized). Is this a mistake? The paper is not reproducible as it stands.

**Reasons To Accept:**

1. The paper addresses the challenge of preserving bias-independent information while neutralizing subjective bias. By incorporating a cycle-consistent network, the proposed approach ensures that the bias-independent content of the original text is preserved in the generated neutral text. This is an essential strength as it maintains the integrity and meaning of the original text. Also, the application of the method towards bias neutralization is novel and exciting to follow up on.

2. In total, the paper proposes an application of a model in a new domain and setting, introduces an auxiliary guidance mechanism that highlights sequences of bias-inducing words, and evaluates the results thoroughly.



**Reasons To Reject:**

1. The authors do not compare the performance of the FairBalance model with many other state-of-the-art models for subjective bias neutralization. Including such comparisons would provide a better understanding of the model's effectiveness and potential improvements. I feel it would have made the model easier to compare to the existing (and more limited) models, such as by Liu, Madanagopal, and Pryzant.

2. The paper has some wording issues. For example, I don't favor word choices such as "the models returned satisfactory results", "both models had a good balance", extensive experiments, or "the best way to evaluate the performance (...) is to conduct unbiased human evaluations". Similar wordings happen occasionally throughout the paper, and I feel descriptions should be specific and the judgment left to the reader.

3. Bias neutralization has ethical implications. However, the paper does not provide any ethics statement, even though such a statement does not affect the page limits. Not providing one is a major drawback.



**Reproducibility:**

2: Would be hard pressed to reproduce the results. The contribution depends on data that are simply not available outside the author's institution or consortium; not enough details are provided.

**Reviewer Confidence:**

3: Pretty sure, but there's a chance I missed something. Although I have a good feel for this area in general, I did not carefully check the paper's details, e.g., the math, experimental design, or novelty.

**Typos Grammar Style And Presentation Improvements:**

Line 920, the sentence "We replaced all proper nouns with generic names ," has an unnecessary space before the comma.
Line 505, the sentence: Since more than  30% of the biased words in our corpus were adjectives, the method of deleting biased words proved more effective by eliminating them and neutralizing the sentences. It is not entirely clear that "them" means adjectives.
Some sources have issues, e.g., the paper in line 711 by Shikha Bordia and Samuel R Bowman is published by now. See https://aclanthology.org/N19-3002/. Sources should be thoroughly checked.
The authors seem to not have added DOIs to the references. I'm not fully aware whether this is an issue of the document to review only, but if they are not added, they need to be.

---

> ### Author Rebuttal · Authors · 2023-08-28
>
> Thanks for the detailed feedback. We have added comments to both "Reasons to Reject" and "Questions to Author". The questions and comments are highlighted in bold and the response following it.
>
> **The authors do not compare the performance of the FairBalance model with many other state-of-the-art models for subjective bias neutralization. Including such comparisons would provide a better understanding of the model's effectiveness and potential improvements. I feel it would have made the model easier to compare to the existing (and more limited) models, such as by Liu, Madanagopal, and Pryzant.**
>
> Pryzant's Join embedding model (Pryzant et al., 2019) and Madanagopal's Reinforced sequence training models (Madanagopal and Caverlee, 2022)  were both supervised training models. However, due to the nature of our dataset being non-parallel, we were unable to train these models using our dataset as they require parallel data. Instead, we trained Pryzant's model and Madanagopal's model using their respective datasets and subsequently evaluated their performance against our dataset.
>
> |             | Neutrality | BERTScore | PPL   |
> |-------------|------------|-----------|-------|
> | Pryzant     | 48.79      |     81.54 | 33.52 |
> | Madanagopal | 67.83      |     92.11 | 24.19 |
> | FairBalance | 69.78      |     96.42 | 26.48 |
>
> Pryzant et al.'s model was primarily designed for single word bias neutralization, and its limitations are evident in its lower bias neutralization score of 48.79 as 70% of the data points in our dataset have multi-word bias. We also observed that Pryzant’s joint embedding model mostly addresses the first instance of bias in a sentence and retains most of the other bias instances following it. In contrast, the reinforced sequence trained models by Madanagopal et al. demonstrated effective performance in bias neutralization, achieving a score of 67.83, which is comparable to our proposed FairBalance model. Notably, the reinforced model also exhibited a better perplexity (PPL) score than our FairBalance model. However, when it comes to content preservation, the FairBalance model outperformed the reinforced sequence trained models significantly. These relative close values in performance might be attributed to the common domain of Wikipedia across these models. It's important to highlight that our FairBalance method offers flexibility due to its ability to operate with non-parallel datasets, enabling its application in diverse domains without the need for parallel data.
>
> ---------------------------------------
>
> **The paper has some wording issues. For example, I don't favor word choices such as "the models returned satisfactory results", "both models had a good balance", extensive experiments, or "the best way to evaluate the performance (...) is to conduct unbiased human evaluations". Similar wordings happen occasionally throughout the paper, and I feel descriptions should be specific and the judgment left to the reader.**
>
> We appreciate your comments on using a neutral tone and letting the reader form their own judgments based on the presented results. In response to your comments, we have made several changes. Following are some examples:
>
> (Before) In the absence of non-parallel data, the best way to evaluate the performance of our neutralization models is to conduct unbiased human evaluations
>
> (Now) In the absence of non-parallel data, we chose to evaluate the performance of our neutralization models through human evaluations to ensure unbiased feedback.
> ____________
>
> (Before) Both the CycleGAN and FairBalance models had a good balance of neutralized input text and preserving semantic content from the original input text.
>
> (Now) Both the CycleGAN and FairBalance models aimed to balance neutralizing the input text with retaining portions of the original semantic content
>
> _____________
>
> (Before) Extensive experiments demonstrate how FairBalance significantly improves subjective bias neutralization compared to other methods.
>
> (Now) In our evaluations involving seven models consisting of adversarial and non-adversarial models, the FairBalance method showed a notable improvement in bias neutralization based on subjective human judgment when compared to other techniques.
>
> __________
>
> (Before) Through extensive experiments, we demonstrate that the proposed approach outperforms conventional text style transfer techniques through a combination of subjective and objective evaluation metrics.
>
> (Now) Through seven models (3 baselines and 4 adversarial) along with human judgment evaluation, we demonstrate that the proposed approach outperforms conventional text style transfer techniques through a combination of subjective and objective evaluation metrics
>
> _____________
>
> **Bias neutralization has ethical implications. However, the paper does not provide any ethics statement, even though such a statement does not affect the page limits. Not providing one is a major drawback.**
>
> Thank you for highlighting the absence of an ethics statement in our study. Initially, we presumed that testing the models in the Wikipedia domain wouldn't pose sensitivity concerns. However, considering the broader implications you've aptly brought to our attention, we have now incorporated an ethics statement into our paper.
> “In our exploration of machine learning models for neutralizing linguistic bias, we acknowledge and emphasize the inherent limitations and challenges. Specifically, our FairBalance model bases its analysis on individual sentences without accounting for the broader context in which they appear. This absence of context might lead to instances where the model misidentifies a sentence as either neutral or biased. We recognize that this potential inaccuracy could influence its operational utility. Readers and practitioners should be cautious and considerate of this limitation when interpreting or deploying the results from our model.”
>
> ___________
>
> **Questions For The Authors:**
> **There is no information about human judges, even though that might be very relevant. How was their education, age, etc.? This information is increasingly included in recent works about bias and relates to their labels.**
>
> A total of 10 judges were used for our human judgment and the details are given below:
> 4 Undergraduate students (under 20)
> 1 non-native English speaker
> 2 Linguistic Experts
> 5-10 years’ experience (master’s degree)
> 15-20 years’ experience (PhD in Computer Science)
> 4 people (30-50+ years)
> 1 Person (30-40 years) (master’s degree)
> 2 people (40-50 years) (1 master’s Degree and 1 PhD)
> 1 Person (50+) (bachelor’s degree)
> A set of 50 biased sentences were selected and processed using the experimented GAN models. This combination of biased and neutral sentences was used for human evaluation by a panel of 10 judges. Every time a biased sentence and its corresponding neutralized sentence from one of the models is presented to the user and asked to rate the following three aspects separately: Neutrality, Content Preservation and Fluency
>
> **The study focuses on specific forms of subjective bias observed at the sentence level. However, considering the context (paragraph) in which a statement is made could potentially alter the perspective. You say one sentence on this, but do you already have ideas on how it will be implemented in your work?**
> Thank you for highlighting the potential influence of context on bias detection and neutralization. While the current bias neutralization model operates solely on biased sentences, broadening the scope to include contextual factors like article titles, section headers, paragraphs, and neighboring figures could significantly enhance its robustness. Even the presence or absence of references could provide valuable insights. Specifically, we can use an additional auxiliary signal for highlighting a sentence in the paragraph that is biased along with detailed biased words in the sentence. Our next research phase will involve incorporating these expanded contextual elements to perform bias neutralization, which we believe will contribute to a more comprehensive understanding of how contextual cues influence bias mitigation.
>
> **The referenced bias-neutralizing papers are not always completely clear in which kind of bias they deal with. However, bias (from linguistic bias to sentiment bias and bias by word choice and other concepts) is very complex. Can you clarify the definition of bias that you apply?**
> We appreciate your insightful question that prompts us to provide a clear definition of the bias we focus on in our research. The type of bias that is focused on in our paper is linguistic bias, which is a type of judgmental language used in presenting facts that are intended to persuade, argue, or otherwise present an opinion. Primarily, we focused on three types of linguistic bias as identified by Recasens et. al.
> Framing bias is an explicit form of bias that reveals the authors’ stance on a topic using one-sided or subjective words. Framing bias is particularly concerned with highlighting a topic in a positive or negative manner. It’s mostly seen in argument situations, where the speaker takes one side and expresses an opinion strongly opposing it or supporting it.
> Epistemological bias is an implicit and a subtle form of bias that tends to cast a doubt in the expressed information. Epistemological bias is linguistically realized using factive verbs, entailments, assertive or hedges. Identifying epistemological bias can be extremely difficult because if the expressed uncertainty in the statement is universally accepted, then the fact is unbiased.
> Demographic Bias is defined as a systematic asymmetry in word choice that reflects the authors’ social-categorical belief towards a described group or individuals on a topic. It roots from the common criticism of Wikipedia is that it is a male-dominated community. As a result, certain controversial topics like marriage, feminism, and abortion may display gender bias.
>
> **There seems to be no link to all data and code (anonymized). Is this a mistake? The paper is not reproducible as it stands.**
> One of the libraries that is used in our code has a dependency on proprietary modules that’s why our code is not shareable at its current state. However, we intend to address this by actively working on eliminating that dependency and making the code publicly available in the future. In the revised version, we will include the dataset utilized for training the discriminator, bias tagger and neutralization components. Open review is not allowing us to add attachments to the submission. To enhance reproducibility, we've also appended additional implementation details in the appendix.
>
> We already made changes to the paper, but open review doesn't allow us to upload the revised version.

---

### Meta-Review · Area_Chair_Sv8r · 2023-09-08

**Recommendation:** 4

**Metareview:**

This paper introduces a new method for reducing subjective bias in text without the need for parallel data using a cycle-consistent adversarial network. The methodology proposed is novel and the method is both conceptually useful and demonstrates an improved performance over prior methods. The paper is well-written. There are minor points that could have been improved -- the method was only trained on Wikipedia; hence, it is unclear how it would fair on other domains. The method has though been tested on a variety of domains. Moreover, the reviewers raise the point that the baselines that the authors compare to are quite old. This is though because there are no more recent baseline methods which operate on non-parallel data. It does make it easier to propose a method which outperforms prior work though.

---

### Decision · Program_Chairs · 2023-10-07

**Decision:**

Accept-Main

**Comment:**

This paper introduces a new method for reducing subjective bias in text without the need for parallel data using a cycle-consistent adversarial network. The methodology proposed is novel and the method is both conceptually useful and demonstrates an improved performance over prior methods. The paper is well-written. There are minor points that could have been improved -- the method was only trained on Wikipedia; hence, it is unclear how it would fair on other domains. The method has though been tested on a variety of domains. Moreover, the reviewers raise the point that the baselines that the authors compare to are quite old. This is though because there are no more recent baseline methods which operate on non-parallel data. It does make it easier to propose a method which outperforms prior work though.